# A Measurement Model and Empirical Analysis of the Coordinated Development of Rural E-Commerce Logistics and Agricultural Modernization

**Zhiqiang Liu [1], Shitong Jia [1], Zixing Wang [2], Caiyun Guo [1],\* and Yanqi Niu [1]**

[1] School of Management Engineering and Business, Hebei University of Engineering, Handan 056038, China
[2] School of Economics, University of Bristol, Bristol BS8 1QU, UK
\* Correspondence: guocaiyun@hebeu.edu.cn

**Abstract:** With technological revolutions and new rural revitalization strategies, rural e-commerce logistics and agricultural modernization are becoming the focus of society. The purpose of the current study is to measure the coupling coordination state of regional rural e-commerce logistics and agricultural modernization, to analyze the obstacles to coordinated development, to forecast the trend, and to propose policy suggestions for coordinated development. Based on the theory of system coordinated development, this paper selects representative indicators reflecting regional rural e-commerce logistics and agricultural modernization, and constructs a coordinated development measurement index system. This paper also constructs a coupling coordination-obstacle degree model, adopting Handan City, Hebei Province as an example, using the data obtained from 2010 to 2019 to measure the coordinated development status of rural e-commerce logistics and agricultural modernization; and analyzes the obstacles affecting coordinated development. By introducing the grey prediction GM (1,1) model and by applying it to the research, the results show that the model is feasible, and the coordinated development level of rural e-commerce logistics and agricultural modernization in Handan City presents a fluctuating upward trend. The prediction results show that the coordinated development of rural e-commerce logistics and agricultural modernization will achieve good coordination in 2024, and relevant development suggestions are proposed. The research can provide some decision-making references for establishing the measurement model, an obstacle-factor analysis, and a trend prediction for the coordinated development of regional rural e-commerce logistics and agricultural modernization.

**Keywords:** e-commerce logistics; agricultural modernization; coupling coordination-obstacle degree model; grey prediction

## 1. Introduction

In recent years, the technological revolution, characterized by considerable investment in capital and technology has completely altered the economic development model. The digital economy driven by e-commerce, has rapidly increased, accelerating the reshaping of the agricultural industry and logistics supply chains. The development of the e-commerce industry has resulted in greater development prospects for the logistics industry and agriculture. In 2020, China's rural, online, retail sales reached CNY 1790 billion, an increase of 8.9% from the previous year, accounting for 15% of the national, online retail sales. The agricultural products network retail sales being CNY 415.89 billion, was an increase of 26.2%. Farmer groups that trade products with supermarkets turned to the e-commerce market during the COVID-19 pandemic [1]. The rural e-commerce and rural logistics industry has the advantages of a lack of direct contact, the precise docking of supply and demand, a close combination of production and marketing, and the promotion of the construction of agricultural modernization. The transformation of digital and information-based agricultural forms has resulted in the reconstruction of the business model of the rural big

market and alleviated the impact of the pandemic on the rural logistics industry to a certain extent. The interaction between e-commerce logistics and agricultural modernization has promoted the integration of rural industries and the benign interactive development of the region. At the same time, the revolution of information technology will also result in a shortage of resources, environmental degradation, and many other factors, affecting the quality and efficiency of both the logistics industry and agriculture. In order to cope with the new challenges imposed by the coordinated development of rural e-commerce logistics and agricultural modernization in the post- pandemic era, it has become an urgent requirement to systematically analyze the coordinated development status and trend of rural e-commerce logistics and agricultural modernization and to propose development countermeasures.

Following the extensive attention given to the issue of "agriculture, rural areas and farmers", and the continuous strengthening of rural construction, in 2015, the Ministry of Agriculture and other departments of China jointly issued the "Several Opinions on Promoting the Healthy Development of Rural Logistics and Accelerating the Service of Agricultural Modernization" to promote the coordinated development of China's rural logistics and agricultural modernization process, comprehensively improving the level of China's rural logistics and supporting the development of agricultural modernization. In 2021, the State Council of China issued the "Opinions on Comprehensively Promoting Rural Revitalization and Accelerating Agricultural and Rural Modernization", encouraging the development of rural e-commerce, building rural logistics outlets, comprehensively promoting rural revitalization, and both accelerating agricultural and rural modernization. In the same year, the "14th Five-Year Plan for Promoting Agricultural and Rural Modernization" issued by the State Council of China highlighted that the commercial service network should be improved, a logistics operation service model should be created, and agricultural modernization should be promoted. Therefore, the development of rural logistics and agricultural modernization has become the only means by which to implement the strategy of rural revitalization in the new era and to realize the innovation of the green, and sustainable development of the regional economy. The current study aims to provide theoretical and methodological suggestions for the coupling and coordination of regional rural e-commerce logistics and agricultural modernization, and to promote the sustainable development of the regional economy.

The ideas presented in this study are as follows: the second section organizes the existing literature and highlights the shortcomings presented in the current research; the third section presents the research design, constructing the measurement index system, coupling coordination-obstacle degree model, and grey prediction GM (1,1) model of the coordinated development of regional rural e-commerce logistics and agricultural modernization; the fourth section focuses on Handan City, Hebei Province, as the case study, and proposes several development proposals; and, finally, the last section presents the conclusion.

## 2. Literature Review

### 2.1. Rural E-Commerce Logistics and Agricultural Modernization

Rural e-commerce logistics is a combination of e-commerce and logistics in rural areas, one of the businesses is agricultural logistics. E-commerce logistics includes urban and rural e-commerce logistics, which is a kind of logistics service. The service object of e-commerce logistics in rural is rural residents, including the flow of agricultural products to the city and commodities to the countryside through the Internet, enterprises manage the logistics operation process through computer network technology to achieve resource sharing between upstream and downstream of the supply chain. Agricultural logistics is the physical flow of agriculture-related material entities and related information from producers to consumers in order to meet the needs of agricultural production and management entities. That is, rural e-commerce logistics transforms the traditional rural logistics industry through the Internet and information-processing technology, and use modern network technology to participate in rural logistics operations and management, to achieve



the exchange of agriculture inter-enterprise logistics information and the scientific allocation of resources. Rural e-commerce logistics provides transportation, warehousing and distribution, information processing, and other services for rural residents' produce, life, and other economic activities [2]. Increasingly, more users use B2C e-commerce channels to purchase goods and send them to their homes through express-delivery companies, which produces more stringent tests for rural logistics [3].

Agricultural modernization refers to the process of integrating traditional agriculture with low production efficiency with modern and technology industries, and introducing modern management methods into agricultural production. It is both a process of transforming agriculture and a means of improving the rural economy. Schultz [4] proposed the theory of transforming traditional agriculture, and highlighted that agricultural modernization is the process of transforming traditional agriculture into modern agriculture. The construction of agricultural modernization is the key to realizing the strategy of rural revitalization, the synchronous development of "mechanization, science, water conservancy, and electrification", and the comprehensive construction of socialist modernization [5,6].

The previously mentioned scholars were mainly selected from the perspective of rural e-commerce logistics or agricultural modernization to analyze a particular aspect, but were less involved in the relationship between comprehensive capacity and development. It is worth noting that rural e-commerce logistics and agricultural modernization are important methods used to revitalize the rural economy. In comparison to the traditional, rural, economic construction model, there is a certain relationship evident between the two in terms of development ideas and results analysis. From the perspective of goal setting, both can take into account the problem of rural employment and the reasonable demands of rural residents. From the perspective of the implementation process, both factors emphasize the preparation of the industrial foundation and interest division of the industrial supply subject in the rural area.

### 2.2. The Relationship between Rural E-Commerce Logistics and Agricultural Modernization

Rural e-commerce logistics supports and guides the process of agricultural modernization. Agricultural modernization has benefited from the efficient and rapid movement of goods and services caused by the logistics industry cluster [7]. The e-commerce logistics industry cluster has created many opportunities for expanding rural logistics enterprises and innovating the economic development model. At the same time, the logistics industry, to successfully accelerate the spatial reorganization of agriculture, mixed agricultural factors to give way to professional production, guide the transformation and upgrading of traditional agriculture to industrial agriculture, and promote the improvement of the rural-market system [8,9]. As an emerging industry, rural e-commerce logistics has been widely analyzed and studied in the academic circle for its role in the development of agricultural modernization. The development of e-commerce logistics has promoted the construction of the agricultural management service system; accelerated the development of agricultural mechanization; promoted the integration of the Internet, agriculture, and logistics industries; promoted the construction of rural organization and socialization; and implemented agricultural modernization [10–14]. The prosperous logistics industry is more likely to attract the government's financial investment in transportation, human settlements, and information to promote the connection between local agricultural economic entities, which has brought more space for the development of agricultural modernization. When the transportation costs of rural enterprises is low, the degree of industrial agglomeration is high [15,16], which lays the foundation for the process of agricultural industrialization. The government subsidizes the sales of agricultural products through e-commerce channels to support the development of local agriculture and to promote China's agricultural development [17]. The above-mentioned scholars made a preliminary exploration of rural e-commerce logistics to promote the development of agricultural modernization and achieved some research results. However, the scholars only analyzed from a single

perspective and lacked systematic research on the interaction between rural e-commerce logistics and agricultural modernization.

Agricultural modernization creates a favorable environment for the development of rural e-commerce logistics. The main reason for the transportation loss of agricultural products is the use of backward vehicles and technologies on low-quality roads, which increases the transportation burden [18]. Due to the limited availability of refrigerated transportation equipment, the excessive demand for e-commerce logistics causes a shortage of rural resources [19]. Agricultural modernization improves local transportation rates and introduces various production technologies, and farmers use mechanical equipment instead of performing manual labor, increasing labor efficiency and reducing the transportation costs of agricultural products [20–22]. Therefore, the development of agricultural modernization can promote the improvement of logistics efficiency to a certain extent. Road and information constructions have resulted in the efficiency of rural and urban–rural trade, connected rural and urban areas, and mobilized circulation of agricultural products [23,24]. At the same time, with the process of agricultural modernization, scientific production technology, and the application of modern science and technology to optimize the layout of agricultural enterprises and logistics processes [25], when agricultural conditions reach a certain level, they cause processing and manufacturing enterprises and e-commerce logistics enterprises to assemble, realize the scale of the industrial economy, and improve the competitiveness of rural e-commerce logistics and modern agriculture. From the above research literature, it can be concluded that agricultural modernization has an important role in promoting the development of the rural e-commerce logistics industry and the transformation of agricultural production models. Most of the above-mentioned scholars demonstrated the one-way interaction between rural e-commerce logistics and agricultural modernization from a single perspective, rarely involving the analysis of the coupling and coordination between rural e-commerce logistics and agricultural modernization.

*2.3. The Measurement of Rural E-Commerce Logistics and Agricultural Modernization*

For the measurement of the agricultural modernization development level, Rong and Wei [26] established the agricultural evaluation index system according to the three dimensions of management scale, basic conditions, and sustainable development. Martinho [27] proposed the need to use numbers of livestock and agricultural imports and exports to reflect the scale of agricultural development. It provides new ideas for index selection. Shu and Hu [28] chose infrastructure level, agricultural scale, and agricultural quality as the measurement dimensions. Konefal et al. [29] provided the measurement index of agriculture from the four following aspects: population, economy, ecology, and agriculture. Zhang et al. [30] added management and industrial systems to the measurement dimension of agricultural modernization. The scholar's research was based on the background of the information technology revolution and industrialization. Jiang and Hu [31] presented agricultural modernization from the perspective of agricultural economy, society, and ecological modernization. In addition, Yoon et al. [32] used the evaporation stress index as a drought assessment indicator to reflect regional resilience. The scholar considered the development of agricultural modernization under special circumstances. One of the characteristics of agricultural modernization is the sustainable development of agriculture. Valizadeh and Hayati [33] included soil health and biophysical compatibility in the measurement system. It has enriched the index selection under the dimension of sustainable development and provided a reference for the subsequent research of scholars.

For the measurement of the development level of e-commerce logistics, Kucukaltan et al. [34] established a list of logistics indicators from the four following perspectives: finance, internal processes, and stakeholders. The index system of logistics established by Liang et al. [35] consisted of logistics scale, input, efficiency, and infrastructure. Martí et al. [36] utilized logistics infrastructure, international transportation channels, logistics quality, traceability, and timeliness of cargo transportation as evaluation indicators of logistics systems. Although the dimensions selected by the above scholars were different,

they were all based on the input–output principle. These scholars presented more research on the construction of rural e-commerce logistics or the agricultural modernization evaluation index system, which is less involved in the construction of rural e-commerce logistics and the agricultural modernization composite system of the coordinated development measurement index system.

### 2.4. The Model of the System Coordinated Development Relationship

At present, with the expansion of industrial economic research and the development of empirical tests, increasingly more scholars use the coupling coordination model to analyze whether there is a coordinated development relationship between agricultural modernization and the related fields. For example, Du et al. [37] studied the coordinated development relationship between agricultural modernization and informatization through the coupling coordination model; Ou et al. [38] quantitatively analyzed and evaluated the coordinated development of new industrialization, informatization, urbanization, agricultural modernization, and greening in Hunan Province. Liu and Yu [39] demonstrated the interaction between agricultural modernization and socialization, where agricultural modernization and socialization promote each other; Chen [40] studied the coordinated development of "five modernizations"; Cai et al. [41] determined that there was an internal relationship between agricultural modernization and rural ecological environment protection; Tian et al. [42] demonstrated the interdependence of agricultural modernization with industrialization, urbanization, and ecosystem services; Liu et al. [43] showed that agricultural modernization promoted urban–rural integration, and urban–rural integration accelerates the process of agricultural modernization; and Liu et al. [44] observe that agricultural modernization and farmers' professionalization are closely linked, and there is a coordinated development relationship between the two. The above researchers prove that the coupling coordination model is suitable for analyzing the coordination relationship between systems.

Meanwhile, in the context of the prosperity and development of the e-commerce industry, the cooperation between e-commerce enterprises and rural logistics is increasingly close, and the development of rural e-commerce logistics is accelerating. Some scholars focused on the coordinated development relationship between the logistics industry and other industries, for example, the research of Zhang and Zhao [45] determined that agricultural e-commerce logistics and agricultural economy are closely linked, and there is a strong cooperative development relationship; Chen et al. [46] studied the mutual influence and support between the logistics industry and national economy; and Yan et al. [47] thoroughly discussed the interaction between logistics and manufacturing industries. With the advent of the Internet economy era, the coupled relationship between rural e-commerce logistics and agricultural modernization has been continuously enhanced. Often, the existing research solely focused on the system coupling coordination relationship, did not explore its development-obstacle factors, and lacked the prediction research for the coordinated development trend of the two; the development proposals presented lacked data support.

Hence, the current study combined entropy method, coupling coordination-obstacle degree model, and grey prediction model to measure rural e-commerce logistics and agricultural modernization, and proposed policy suggestions based on the research results. In the current study, the grey prediction model was used to predict the coordination trend between systems. The original data were from the calculation results of the coupling coordination-obstacle degree model, and the prediction results are relatively accurate. The coupling coordination-obstacle degree model could not only analyze the comprehensive development level of the system, test the coordination state, but also identify the main obstacle-factors affecting the coordinated development of the system. In the process of modeling the coupling coordination-obstacle model, it was necessary to calculate the weight of the evaluation index, which was calculated by the entropy method. Compared with the subjective allocation method, the entropy method is more accurate and objective. The

combination of entropy method, coupling coordination-obstacle degree model, and grey prediction model can provide data support for the coordinated development suggestions.

Based on the theoretical framework of the existing research, the current study constructed the measurement index system for the coordinated development of rural e-commerce logistics and agricultural modernization from the perspective of coordinated development, established the coupling coordination-obstacle degree model, introduced the grey prediction GM (1,1) model, and applied it to quantitatively predict the coordinated development trend of the two. Thereafter, it proposed the development countermeasures and suggestions according to the relevant analysis results, which can provide theoretical and methodological support for the coordinated development of rural e-commerce logistics and agricultural modernization.

## 3. Research Design

### 3.1. The Construction of the Measurement Index System of Coordinated Development

Constructing a scientific and reasonable measurement index system for the coordinated development of rural e-commerce logistics and agricultural modernization is the premise and key to analyzing the coupling and coordination relationship between the two, as well as realizing the coordinated development. Referring to the relevant research results [36–46], the rural e-commerce logistics system includes four dimensions: logistics infrastructure, cost, efficiency, and scale. The agricultural modernization system also includes four dimensions, namely, agricultural cost, efficiency, sustainable development, and social development. The logistics infrastructure provides the foundations for information and hardware for the cross-integration of rural e-commerce logistics and agricultural modernization. The logistics' cost, efficiency, and scale comprehensively summarize the popularity of e-commerce logistics in rural areas. One of the characteristics of agricultural modernization is to realize the sustainable development of agriculture. Agricultural cost determines the upper limit of agricultural modernization. Therefore, agricultural cost, efficiency, and sustainable development represent the development level of agricultural modernization. An agricultural society reflects the consumption ability of rural residents and objectively reflects the development prospect of e-commerce logistics in rural areas. Considering domestic and foreign scholars in the construction of rural e-commerce logistics and the agricultural modernization coordinated development measure index system, there exists a considerable difference in the choice of dimension; the index-selection difference is minor. Therefore, starting from the concept of coordinated development, based on the principles of system, simplicity, and data availability, and considering the development of information technology and agricultural industrialization, this study focused on the selection of indicators, such as rural mobile-phone users and the agricultural industrialization operation rate, and constructed a measurement index system for the coordinated development of rural e-commerce logistics and agricultural modernization from the system and index levels, as presented in Table 1.

### 3.2. The Methods of Research

#### 3.2.1. Original Data Standardized Processing

The original data were processed into dimensionless values by the standardization method, which was convenient for performing a comparison between indicators of different units or orders of magnitude. Because in the subsequent calculation, standardized index as a denominator cannot be 0, in order to avoid the datum of a standardized index obtaining a value of 0 and affecting the calculation result, and in order to maintain the significance of the index and the fairness between the indicators, the standardized value of each index was increased by 0.01.

**Table 1.** Measurement index system of coordinated development of rural e-commerce logistics and agricultural modernization.

| System | Index | Calculation Method (Data Source) | Attribute | Weight |
|---|---|---|---|---|
| Rural e-commerce logistics | Rural freight volume $Y_1$ (million tons) | Regional freight volume * rural road mileage ratio | positive | 11.36% |
| | Fixed assets investment in transportation, storage and postal industries $Y_2$ (million yuan) | Access to statistics | positive | 11.38% |
| | rural highway mileage $Y_3$ (km) | Access to statistics | positive | 10.25% |
| | Rural mobile phone users $Y_4$ (household) | Regional total mobile phone users * rural population proportion | positive | 6.00% |
| | Rural internet broadband access users $Y_5$ (household) | Regional network broadband access households * rural population proportion | positive | 10.06% |
| | Rural logistics practitioners $Y_6$ (person) | Total regional logistics industry * proportion of rural population | positive | 5.43% |
| | Total rural post and telecommunication business $Y_7$ (billion yuan) | Regional total post and telecommunication business * rural population proportion | positive | 24.88% |
| | Total rural foreign-trade exports $Y_8$ (millions of dollars) | Regional total foreign trade exports * rural population proportion | positive | 6.19% |
| | Total rural foreign-trade imports $Y_9$ (millions of dollars) | Regional total foreign trade imports * rural population proportion | positive | 14.46% |
| Agricultural modernization | Total power of agricultural machinery per unit area $X_1$ (kW/ha) | Regional agricultural machinery total power/total planting area | positive | 16.16% |
| | Effective water duty $X_2$ (%) | Effective irrigation area/arable land area | positive | 9.36% |
| | Fertilizer application rate per unit area $X_3$ (tonnes/ha) | Regional fertilizer application rate/total seeding area | positive | 11.32% |
| | Rural electricity consumption $X_4$ (millions of kWh) | Access to statistics | positive | 13.13% |
| | Per capita output of grain $X_5$ (kg/person) | Regional total grain output/total population | positive | 7.89% |
| | Rural per capita net income $X_6$ (yuan) | Access to statistics | positive | 12.10% |
| | Urban-rural consumption ratio $X_7$ (%) | Access to statistics | negative | 10.99% |
| | Agricultural industrialization rate $X_8$ (%) | Access to statistics | positive | 12.28% |
| | Agricultural disaster rate $X_9$ (%) | Regional agricultural disaster area/agricultural disaster area | negative | 6.76% |

For the positive indicators, the greater the index value, the greater the impact on the system. The calculation formula is as follows:

$$Z_{ij} = \frac{\left[x_{ij} - \min(x_{ij})\right]}{\left[\max(x_{ij}) - \min(x_{ij})\right]} + 0.01 \tag{1}$$

For the negative indicators, the smaller the index value, the greater the impact on the system. The calculation formula is as follows:

$$Z_{ij} = \frac{\left[\max(x_{ij}) - x_{ij}\right]}{\left[\max(x_{ij}) - \min(x_{ij})\right]} + 0.01 \tag{2}$$

where $x_{ij}$ represents the jth data of the i index. $Z_{ij}$ represents the data of each original index following standardization, $\max(x_{ij})$ represents the upper limit of the index, and $\min(x_{ij})$ represents the lower limit of the index.

### 3.2.2. Constructing of the Coupling Coordination-Obstacle Degree Model

In this paper, the compound system of rural e-commerce logistics and agricultural modernization was defined as $S = \{S_1, S_2\}$, the subsystem of rural e-commerce logistics was $S_1$, and the subsystem of agricultural modernization was $S_2$.

(1)    Calculation of the index weight by the entropy method

The entropy method is an objective weighting method that calculates the index weight according to the variation degree of the index value, avoids the deviation caused by subjective factors, and has greater accuracy and objectivity than the subjective weighting method. The greater the degree of variation of an indicator, the less the information entropy;

the greater the amount of information provided by the indicator, the greater the weight of the indicator should be. When the index value variation degree of an index is smaller, the information entropy is larger, the information provided by the index is smaller, and the weight of the index is smaller. Therefore, according to the degree of variation of each index value, the information entropy can be used to calculate the index weight and provide the basis for the comprehensive evaluation of the index [48].

$$P_{ij} = \frac{Z_{ij}}{\sum_{i,j=1}^{n} Z_{ij}} \tag{3}$$

$$e_i = -\frac{1}{\ln(n)} * \sum_{i,j=1}^{n} \left(P_{ij} * \ln\left(P_{ij}\right)\right) \left(0 \leq e_j \leq 1\right) \tag{4}$$

$$\Omega_i = \frac{1 - e_i}{\sum_{j=1}^{n} 1 - e_i} \tag{5}$$

where $P_{ij}$ represents the proportion of indicators, $Z_{ij}$ represents the standardized data of each original indicator, $e_i$ represents the information entropy value of the index, and $\Omega_i$ represents the index weight.

(2)    Calculation of system comprehensive development index

On the basis of the weight calculated by the entropy method, a comprehensive evaluation model was constructed to evaluate the comprehensive development level of the rural e-commerce logistics and agricultural modernization system in the regional composite system. The comprehensive development index of each subsystem was calculated to reflect the comprehensive development status between the systems.

$$H_y = \sum_{i,j=1}^{n} Z_{ij} * \Omega_i, H_x = \sum_{i,j=1}^{n} Z_{ij} * \Omega_i \tag{6}$$

where $H_y$ is used to represent the comprehensive development index of rural e-commerce logistics, and $H_x$ is used to represent the comprehensive development index of agricultural modernization.

(3)    Calculation of system coupling coordination degree

System coupling coordination is mainly manifested as the interaction between the system or subsystem by itself and the outside world. In this paper, the coupling coordination degree model was applied to the research on the coordinated development of regional rural e-commerce logistics and agricultural modernization, which can objectively reflect the coordination degree between the two systems and avoid the influence of subjective factors. The calculation of the coupling coordination degree can reflect the strength of the interaction between systems.

$$C = 2 \times \left[\frac{H_x * H_y}{\left(H_x + H_y\right)^2}\right]^{\frac{1}{2}} \left(0 \leq C \leq 1\right) \tag{7}$$

$$T = \alpha H_x + \beta H_y \left(0 \leq T \leq 1\right) \tag{8}$$

$$D = \sqrt{C * T} \left(0 \leq D \leq 1\right) \tag{9}$$

where C is the coupling degree, which indicates the degree of interaction between the two systems, realizes the dynamic correlation of coordinated development, and can reflect the degree of interdependence and mutual restriction between different systems. T is the coordination index, where $\alpha$ and $\beta$, respectively, represent the undetermined coefficients of rural e-commerce logistics and agricultural modernization in the process of economic development, reflecting the importance of the two in coupling and coordination. The

sum of $\alpha$ and $\beta$ is equal to 1. As a dominant industry in the development of national economy, rural e-commerce logistics and agricultural modernization are important ways to revitalize the rural economy. Therefore, this study set the two as equally important, so both $\alpha$ and $\beta$ were 0.5. D is the coupling coordination degree, which indicates the coordinated development degree of the two systems. The greater the D value, the higher the degree of coordination between the two, the better the coordination state. The division standard of the coupling stage is presented in Table 2, and the division standard of the coupling coordination degree is presented in Table 3.

**Table 2.** Coupling stage classification criteria.

| **Coupling Phase** | **Low Level** | **Average Level** | **Higher Level** | **High Level** |
|---|---|---|---|---|
| | $0 < C \leq 0.3$ | $0.3 < C \leq 0.5$ | $0.5 < C \leq 0.8$ | $0.8 < C \leq 1$ |

**Table 3.** Coupling coordination degree classification criteria.

| **D Interval** | **Coupling Coordination Degree** | **D Interval** | **Coupling Coordination Degree** |
|---|---|---|---|
| $0.0 \leq D < 0.1$ | Extreme disorders | $0.5 \leq D < 0.6$ | Reluctant coordination |
| $0.1 \leq D < 0.2$ | Severe disorders | $0.6 \leq D < 0.7$ | Primary coordination |
| $0.2 \leq D < 0.3$ | Moderate disorders | $0.7 \leq D < 0.8$ | Intermediate coordination |
| $0.3 \leq D < 0.4$ | Mild disorders | $0.8 \leq D < 0.9$ | Good coordination |
| $0.4 \leq D < 0.5$ | Endangered disorders | $0.9 \leq D \leq 1.0$ | Quality coordination |

(4) Calculation of obstacle degree

After calculating the standardized data and index weight, in order to obtain the main obstacle-factors of the coordinated development of rural e-commerce logistics and agricultural modernization, the obstacle degree model was used to study the results, provide data support for the follow-up recommendations to promote the coordinated development of rural e-commerce logistics and agricultural modernization.

$$I_j = 1 - Z_{ij} \tag{10}$$

$$O_{ij} = \frac{\Omega_i * I_j}{\sum_{i=1}^{n} \Omega_i * I_j} * 100\% \tag{11}$$

where $I_j$ represents the index deviation, $Z_{ij}$ represents the standardized data of each original indicator, $O_{ij}$ represents the obstacle degree of index i, indicating the degree of influence of the *j*th data of index i on system development, and $\Omega_i$ denotes the weight of index i.

### 3.2.3. Constructing the Grey Prediction GM (1,1) Model

The grey prediction model can predict the system with uncertain factors. By analyzing the degree of dissimilarity between the development trends of the system indicators, the original index data are generated to obtain the law of system change, and the data sequence with strong regularity is constructed. Then, the corresponding differential equation model is established to predict the future development trend of things [49]. The establishment process of the grey prediction GM (1,1) model is as follows:

(1) Testing of level ratio

The feasibility of the modeling method must be guaranteed before the grey prediction GM (1,1) model is established, that is, the original data need to be tested by the ratio level test [50]. Let the original sequence be $x^{(0)} = \left(x^{(0)}(1), x^{(0)}(2), \ldots, x^{(0)}(m)\right)$; m is the quantity of original data. Perform a ratio test on the original sequence.

$$\lambda(t) = \frac{x^{(0)}(t-1)}{x^{(0)}(t)}, t = 2, 3, \ldots, m \tag{12}$$

where if all $\lambda(t)$ are in the interval $B = \left(e^{\frac{-2}{m+1}}, e^{\frac{2}{m+1}}\right)$, $x^{(0)}$ can establish the GM (1,1) model. Otherwise, the original sequence needs to be shifted, and the constant c is obtained so that the order ratio falls within the interval, that is, $y^{(0)}(t) = x^{(0)}(t) + c$, $t = 1, 2, \ldots, m$.

(2)    Generation of a cumulative sequence

The essence of the grey prediction model is to excavate the evolution situation of the information accumulation process by accumulation operator. The cumulative generation processing is to weaken the randomness and uncertainty of the original sequence, so as to improve the stability and reliability of the grey prediction model [51]. The generated cumulative sequence is as follows:

$$x^{(1)} = \left(x^{(1)}(1), x^{(1)}(2), \ldots, x^{(1)}(m)\right)$$

When $x^{(0)}$ or $y^{(0)}$ pass the level ratio test, an accumulation process is performed.

$$x^{(1)}(t) = \sum_{i=1}^{t} x^{(0)}(i), t = 1, 2, \ldots, m \tag{13}$$

(3)    Construction of GM (1,1) model

According to the cumulative sequence, the grey prediction GM (1,1) model is established.

$$x^{(0)}(t) + \alpha x^{(1)}(t) = \mu \tag{14}$$

The corresponding whitening model is:

$$\frac{dx^{(1)}}{dt} + \alpha x^{(1)}(t) = \mu \tag{15}$$

where $\alpha$ is the development coefficient and $\mu$ is the grey action.

(4)    Generation of a sequence of equal-weighted neighbor values

$$z^{(1)} = \left(z^{(1)}(2), z^{(1)}(3), \ldots, z^{(1)}(t)\right)$$

$$z^{(1)}(t) = \gamma x^{(1)}(t) + (1 - \gamma)x^{(1)}(t - 1), t = 2, 3, \ldots, m \tag{16}$$

where $z^{(1)}(t)$ is the number of adjacent values generated by the cumulative sequence $x^{(1)}$ under the weight $\gamma$. $x^{(1)}(t - 1)$ and $x^{(1)}(t)$ are a pair of neighbor values in the cumulative sequence $x^{(1)}$, $x^{(1)}(t - 1)$ is the posterior neighbor value, and $x^{(1)}(t)$ is the anterior neighbor value. $\gamma$ represents the weight factor, $\gamma \in [0, 1]$. Referring to the existing research, the current paper utilized $\gamma = 0.5$. The grey prediction GM (1,1) model can be transformed into:

$$x^{(0)}(t) + \alpha z^{(1)}(t) = \mu \tag{17}$$

(5)    Calculation of unknown parameters

By using the least-square fitting, we can obtain:

$$\hat{\alpha} = \left(B^T B\right)^{-1} B^T U_n = [\alpha, \mu]^T$$

Calculate the estimated values of parameters $\alpha$ and $\mu$, where B is the data matrix and $U_n$ is the data vector, respectively:

$$
B = \begin{bmatrix} -\frac{1}{2}\left(x^{(1)}(1) + x^{(1)}(2)\right) & 1 \\ -\frac{1}{2}\left(x^{(1)}(2) + x^{(1)}(3)\right) & 1 \\ \vdots & \vdots \\ -\frac{1}{2}\left(x^{(1)}(m-1) + x^{(1)}(m)\right) & 1 \end{bmatrix} \quad U_n = \begin{bmatrix} x^0(2) \\ x^0(3) \\ \vdots \\ x^0(m) \end{bmatrix}
$$

(6) Recovery of predicted values

By substituting the calculated $\alpha$ and $\mu$ values into the grey prediction GM (1,1) model, the solution $\hat{x}^{(1)}(t)$ of the model is calculated and the predicted value $\hat{x}^{(0)}(t)$ is restored.

$$
\hat{x}^{(1)}(t) = \left(x^0(1) - \frac{\mu}{\alpha}\right)e^{-\alpha(t-1)} + \frac{\mu}{\alpha} \tag{18}
$$

$$
\hat{x}^{(1)}(t+1) = (\hat{x}^{(0)}(1) - \frac{\mu}{\alpha})e^{-\alpha t} + \frac{\mu}{\alpha}, t = 1, 2, \ldots, m-1 \tag{19}
$$

$$
\hat{x}^{(0)}(t) = \hat{x}^{(1)}(t+1) - \hat{x}^{(1)}(t), t = 1, 2, \ldots, m-1 \tag{20}
$$

(7) Testing of accuracy level

After the calculation of the grey prediction model is completed, it is necessary to pass the accuracy level test to determine whether the predicted value is reasonable. In this paper, the prediction accuracy was tested by the mean-square-error ratio and small-error-probability value.

$$
S_1^2 = \frac{1}{n}\Sigma_{t=1}^n \left[x^{(0)}(t) - \frac{1}{n}\Sigma_{t=1}^n x^{(0)}(t)\right]^2 \tag{21}
$$

$$
S_2^2 = \frac{1}{n}\Sigma_{t=1}^n \left[e(t) - \frac{1}{n}\Sigma_{t=1}^n e(t)\right]^2 \tag{22}
$$

$$
e(t) = x^{(0)}(t) - \hat{x}^{(0)}(t), t = 1, 2, \ldots, m \tag{23}
$$

$$
K = \frac{S_2}{S_1} \tag{24}
$$

$$
\rho = \left\{e(t) - \frac{1}{n}\Sigma_{t=1}^n e(t) < 0.6745 S_1\right\} \tag{25}
$$

where $S_1^2$ represents the original sequence variance, $S_2^2$ represents the residual sequence variance, and $e(t)$ represents the residual. K represents the ratio of the mean-square deviation and $\rho$ represents the probability of the small error, which is used to measure the fitting accuracy of the model. The classification standard of model accuracy is presented in Table 4. When testing the model, the smaller the K value, the better the $\rho$ value.

**Table 4.** GM (1,1) model accuracy classification criteria.

| | Good | Qualified | Barely Qualified | Unqualified | Good |
|---|---|---|---|---|---|
| **Model Accuracy** | $\rho > 0.95$ | $\rho > 0.8$ | $\rho > 0.7$ | $\rho \leq 0.7$ | $\rho > 0.95$ |
| | $K \leq 0.35$ | $0.35 < K \leq 0.5$ | $0.5 < K \leq 0.65$ | $0.65 < K$ | $K \leq 0.35$ |

## 4. Empirical Analysis

### 4.1. An Overview of the Region

Handan belongs to Hebei Province, China. It is a national logistics hub layout carrying city and has a unique advantage in the development of e-commerce logistics. The total sown area of crops in Handan is 11.71 million mu. It is a large agricultural city in Hebei Province and a pilot "tons of grain market" in China. Agricultural modernization plays an important role in rural revitalization. At the same time, Handan City has six districts, 12 counties, and one county-level city, which is a typical resource-based city. Coal and steel resources have helped its economy steadily increase. At present, the transformation of traditional industries to modern industries is the future development direction. From this perspective, the scientific analysis and prediction of the coordinated development of rural e-commerce logistics and agricultural modernization in Handan City will help to promote the high-quality development of the regional economy.

### 4.2. Data Sources

This paper analyzed the data related to rural e-commerce logistics and agricultural modernization in Handan City, Hebei Province. The original data were derived from the 2010–2019 "China Rural Statistical Yearbook", "Hebei Economic Yearbook", "Handan Statistical Yearbook", and "Handan Statistical Bulletin of National Economic and Social Development", or were obtained through relevant calculation formulas.

### 4.3. The Calculation of Coupling Coordination-Obstacle Degree

Using Formulas (1)–(5), the index weight was calculated; see Table 1. The comprehensive development index, coupling degree and coupling coordination degree of rural e-commerce logistics and agricultural modernization in Handan City from 2010 to 2019 were calculated using Formulas (6)–(9). For example, the 2010–2019 original data for $Y_1$ were, in order, 22,203.93577, 25,539.95767, 30,148.82102, 32,473.06014, 32,807.64824, 36,199.41886, 18,068.7954, 20,005.20349, 22,401.30545, 22,513.47559, and the maximum was 36,199.41886, while the minimum was 18,068.7954. The original data were standardized based on the maximum and minimum, and the processed data were 0.238074913, 0.422074206, 0.676277453, 0.804471562, 0.822925869, 1.01, 0.01, 0.116803172, 0.248960897, 0.255147675, where the sum equaled 4.604735748. Using the sum of the standardized data, $P_{ij}$ and $e_i$ were calculated, where $\ln(10)$ was approximately equal to 2.302585093, $e_i$ was equal to 0.878918767, and "$1 - e_i$" was equal to 0.121081233. Similarly, the "$1 - e_i$" of other indicators under the rural e-commerce logistics system was calculated, the weight of the indicator in the system was found by divided the "$1 - e_i$" of the indicator by the sum of the "$1 - e_i$" of all indicators under the rural e-commerce logistics system. The weight of $Y_1$ was 11.36%. Based on the product of the standardized data of the indicator and the corresponding weight, the sum was the comprehensive development index of the system from 2010 to 2019. The comprehensive development index of rural e-commerce logistics from 2010 to 2019 was taken as 0.281873, 0.301486, 0.417866, 0.474221, 0.468280, 0.485860, 0.388293, 0.361333, 0.498969, and 0.717579. In the same way, the comprehensive development index of agricultural modernization was obtained. Based on the comprehensive development index of rural e-commerce logistics and agricultural modernization, and according to Formulas (7)–(9), C, T, and D values were calculated. Respectively, the calculation results as presented in Table 5.

**Table 5.** Coupling coordination degree of rural e-commerce logistics and agricultural modernization in Handan, 2010–2019.

| Year | $H_y$ | $H_x$ | C | Coupling Phase | D | Coupling Coordination Degree |
|------|-------|-------|---|----------------|---|------------------------------|
| 2010 | 0.281873 | 0.345537 | 0.994838 | High level | 0.558646 | Reluctantly coordinated |
| 2011 | 0.301486 | 0.474274 | 0.974879 | High level | 0.614928 | Primary coordination |
| 2012 | 0.417866 | 0.583538 | 0.986220 | High level | 0.702710 | Intermediate coordination |
| 2013 | 0.474221 | 0.627682 | 0.990255 | High level | 0.738636 | Intermediate coordination |
| 2014 | 0.468280 | 0.699260 | 0.980236 | High level | 0.756460 | Intermediate coordination |
| 2015 | 0.485860 | 0.819035 | 0.966855 | High level | 0.794243 | Intermediate coordination |
| 2016 | 0.388293 | 0.664494 | 0.964972 | High level | 0.712710 | Intermediate coordination |
| 2017 | 0.361333 | 0.731073 | 0.940979 | High level | 0.716914 | Intermediate coordination |
| 2018 | 0.498969 | 0.541555 | 0.999162 | High level | 0.720989 | Intermediate coordination |
| 2019 | 0.717579 | 0.484355 | 0.980993 | High level | 0.767818 | Intermediate coordination |

From Table 5, it can be observed that the coupling-degree C value of the Handan rural e-commerce logistics and agricultural modernization system from 2010 to 2019 remains at a high level, but it cannot represent that both are at a high level of development, nor can it represent that the two are at a high level of coupling and coordination degrees. The analysis was as follows:

(1) The comprehensive development level of the two presented a fluctuating upward trend. The government attaches importance to the impact of rural e-commerce on logistics development and agricultural modernization in rural areas, and introduced policies to encourage and support the development of e-commerce platforms, promoting the overall development of rural e-commerce logistics and agricultural modernization. It can be observed in Table 5 that the comprehensive development index of rural e-commerce logistics in Handan City is 0.281873 in 2010 and increased to 0.717579 in 2019, an increase of 154%. The comprehensive development index of agricultural modernization in Handan also achieved a fluctuating rise, an increase of 40.17% in five years. Although the two reached a high-level coupling stage, the two as a whole still remained at a low level of development. From 2010 to 2018, the comprehensive development level of rural modernization in Handan City was greater than that of rural e-commerce logistics, which belongs to the lagging development of rural e-commerce logistics. After 2018, the comprehensive development index of rural e-commerce logistics in Handan City exceeded agricultural modernization. Rural e-commerce logistics have developed rapidly, while agricultural modernization has lagged behind. The gap between the two has gradually increased over time. COVID-19 broke out in 2019. Affected by the pandemic prevention and control policy, offline farmers' trade activities were limited. The agricultural economy suffered and the construction of agricultural modernization was prevented, resulting in a downward trend in the comprehensive development level of agricultural modernization. Rural e-commerce logistics, with the advantages of online transactions and no direct contact, seized the development opportunity during the outbreak of the pandemic and ushered in the development climax. At the same time, with the increase in the proportion of the rural economy in the gross national product, the government increased its support to rural areas, improved rural infrastructure, encouraged modern logistics talents to actively participate in rural logistics constructions, and established a good foundation for the coordinated development of rural e-commerce logistics and agricultural modernization.

(2) The level of coordinated development between the two still needs to be improved. From 2010 to 2019, the coupling degree between rural e-commerce logistics and agricultural modernization in Handan City remained between 0.8 and 1, maintaining a high level of coupling, indicating that the interaction between the two systems in the past ten years is strong and the impact is deep. A change in any system leads to the instability of the development of another system with a high degree of

coupling. From the D value presented in Table 5, it can be observed that the coupling coordination degree between the two is always in the range of 0.5–1 from 2010 to 2019, and has remained in the coordination stage. However, the D value decreased from 0.794243 in 2015 to 0.71271 in 2016. This was because in 2016, North China was affected by strong convective weather and suffered rainstorm and flood disasters, which led to the destruction of agricultural production in Handan City and the impact of the rural logistics industry. The degree of coupling coordination decreased from good to intermediate. When the COVID-19 pandemic began, rural e-commerce logistics and agricultural modernization were closely cooperating, contributing to solving the shortage of urban agricultural products and maintaining rural economic development. Therefore, the coupling coordination degree increased slightly in 2019. On the whole, there was a stable state of coordinated development between rural e-commerce logistics and agricultural modernization in Handan City from 2016 to 2019. The C value was stable at approximately 0.98, and the D value was stable at approximately 0.75. The degree of coordination between the two was in intermediate coordination and maintained an upward trend. However, rural e-commerce logistics and agricultural modernization have not yet reached the best coordination degree, and are still at a low level of coordination, indicating that although there is a certain coupling and coordination relationship between the two, the degree of coordinated development needs to be further improved.

Therefore, it is necessary to analyze the obstacle-factors affecting the development of rural e-commerce logistics and agricultural modernization in Handan City according to the standardized data, and to propose policy suggestions for the main obstacle-factors in order to achieve the best coordinated development of the two as soon as possible. According to Formulas (10) and (11), the top-five obstacle-factors of rural e-commerce logistics and agricultural modernization development in Handan City from 2010 to 2019 can be calculated. For example, based on the standardized values of the indicators calculated according to Formulas (1) and (2), from 2010–2019, the index deviation for $Y_1$ was 0.761925087, 0.577925794, 0.323722547, 0.195528438, 0.177074131, −0.01, 0.99, 0.883196828, 0.751039103 and 0.744852325 in that order. Through the indicator weight and index deviation in the system, the obstacle degrees of the indicators can be calculated. According to the index obstacle degrees, the ranking of them was carried out, and the top-five obstacle-factors were analyzed per year, as presented in Table 6.

According to Table 6, the obstacle-factors affecting the development of rural e-commerce logistics and agricultural modernization in Handan City have slightly changed. $Y_7$ appeared eight times in ten years, with the frequency of 80%. $Y_2$ and $Y_8$ appeared six times, with the frequency of 60%. This shows that the main obstacle-factors affecting the development of rural e-commerce logistics during this period are the total rural post and telecommunication business and fixed-assets investments in transportation, storage, and postal industries, and total rural foreign-trade exports. $X_6$, $X_8$, $X_5$, and $X_9$ present a high frequency in the ten years, which is greater than 50%. Therefore, the rural per capita net income, agricultural industrialization rate, per capita output of grain, and agricultural disaster rate are the main obstacles affecting the process of agricultural modernization in Handan City.

In the past five years, the frequency of $Y_9$ as the main obstacle-factor was 100%, and the frequency of $X_1$, $Y_1$, and $Y_6$ were 80%. The total rural foreign-trade imports, total rural post and telecommunication business, rural freight volume, and total power of agricultural machinery per unit area have become the main obstacle-factors affecting the coordinated development of rural e-commerce logistics and agricultural modernization in 2015–2019. Therefore, in order to improve the coordinated development level of rural e-commerce logistics and agricultural modernization, it is necessary to improve the infrastructure construction in rural areas, improve the mechanization level of agricultural production, develop rural postal and telecommunications services, encourage the import and export trade of agricultural products, improve the income level of farmers, accelerate the develop-

ment of agricultural industrialization, improve production and farming technology, and improve food production.

**Table 6.** Main obstacle-factors of coordinated development of rural e-commerce logistics and agricultural modernization in Handan, 2010–2019.

| Year | Criteria Scheduling | | | | | | | | | |
|------|--------|-------------------|--------|-------------------|--------|-------------------|--------|-------------------|--------|-------------------|
| | **1** | | **2** | | **3** | | **4** | | **5** | |
| | **Factor** | **Obstacle Degree** | **Factor** | **Obstacle Degree** | **Factor** | **Obstacle Degree** | **Factor** | **Obstacle Degree** | **Factor** | **Obstacle Degree** |
| 2010 | $Y_4$ | 0.2846 | $Y_8$ | 0.2539 | $Y_3$ | 0.2125 | $Y_2$ | 0.1660 | $Y_1$ | 0.1412 |
| | $X_7$ | 0.2476 | $X_5$ | 0.2429 | $X_8$ | 0.2363 | $X_6$ | 0.1999 | $X_4$ | 0.1760 |
| 2011 | $Y_4$ | 0.2075 | $Y_3$ | 0.1988 | $Y_2$ | 0.1735 | $Y_5$ | 0.1709 | $Y_7$ | 0.1299 |
| | $X_8$ | 0.2106 | $X_7$ | 0.2015 | $X_6$ | 0.1721 | $X_4$ | 0.1343 | $X_5$ | 0.1304 |
| 2012 | $Y_3$ | 0.1710 | $Y_5$ | 0.1443 | $Y_4$ | 0.1292 | $Y_7$ | 0.1270 | $Y_2$ | 0.1154 |
| | $X_7$ | 0.1978 | $X_8$ | 0.1721 | $X_6$ | 0.1486 | $X_9$ | 0.1088 | $X_4$ | 0.0777 |
| 2013 | $Y_5$ | 0.1341 | $Y_3$ | 0.1268 | $Y_7$ | 0.1236 | $Y_4$ | 0.1109 | $Y_8$ | 0.0960 |
| | $X_2$ | 0.2194 | $X_8$ | 0.1336 | $X_6$ | 0.1299 | $X_5$ | 0.1062 | $X_9$ | 0.1003 |
| 2014 | $Y_2$ | 0.1467 | $Y_7$ | 0.1215 | $Y_5$ | 0.1205 | $Y_3$ | 0.1058 | $Y_6$ | 0.0708 |
| | $X_5$ | 0.1555 | $X_2$ | 0.1528 | $X_9$ | 0.1108 | $X_6$ | 0.1073 | $X_8$ | 0.0951 |
| 2015 | $Y_9$ | 0.1400 | $Y_5$ | 0.1246 | $Y_7$ | 0.1191 | $Y_4$ | 0.0972 | $Y_2$ | 0.0944 |
| | $X_2$ | 0.1181 | $X_6$ | 0.0919 | $X_8$ | 0.0720 | $X_7$ | 0.0452 | $X_5$ | 0.0415 |
| 2016 | $Y_1$ | 0.1835 | $Y_9$ | 0.1751 | $Y_7$ | 0.1074 | $Y_6$ | 0.1038 | $Y_8$ | 0.1023 |
| | $X_1$ | 0.2227 | $X_2$ | 0.1416 | $X_9$ | 0.0812 | $X_6$ | 0.0731 | $X_8$ | 0.0515 |
| 2017 | $Y_9$ | 0.1879 | $Y_8$ | 0.1786 | $Y_1$ | 0.1637 | $Y_7$ | 0.1134 | $Y_6$ | 0.1121 |
| | $X_1$ | 0.2061 | $X_2$ | 0.1083 | $X_9$ | 0.1017 | $X_6$ | 0.0522 | $X_3$ | 0.0501 |
| 2018 | $Y_9$ | 0.2017 | $Y_6$ | 0.1547 | $Y_8$ | 0.1501 | $Y_1$ | 0.1392 | $Y_2$ | 0.0703 |
| | $X_3$ | 0.4308 | $X_4$ | 0.2550 | $X_1$ | 0.1909 | $X_5$ | 0.0931 | $X_7$ | 0.0797 |
| 2019 | $Y_6$ | 0.2368 | $Y_9$ | 0.1814 | $Y_1$ | 0.1381 | $Y_8$ | 0.0882 | $Y_7$ | −0.001 |
| | $X_3$ | 0.4248 | $X_9$ | 0.2750 | $X_4$ | 0.2440 | $X_1$ | 0.1748 | $X_5$ | 0.1239 |

*4.4. The Grey Prediction of the Coordinated Development Trend*

Based on the grey prediction GM (1,1) model, the current paper dynamically modeled the comprehensive development level and coupling coordination degree of rural e-commerce logistics and agricultural modernization in Handan City from 2020 to 2024, so as to analyze the future development trends of the two. Based on the comprehensive development index and coupling coordination degree of rural e-commerce logistics and agricultural modernization in Handan City from 2010 to 2019, grey prediction was performed. The prediction results all passed the level-ratio and accuracy-level tests, as presented in Table 7 and Figure 1.

When constructing the GM (1,1) model, the testing of the level ratio was first performed to determine whether the data sequence was suitable for the model's construction. The level ratio test values of D were all within the standard range interval (0.834,1.99), indicating that the D data sequence was suitable for the construction of the GM (1,1) model. The level ratio test results for $H_y$ and $H_x$ show that the original data do not pass the level ratio test. Therefore, the translation conversion was performed, and the constant 1 was added on the basis of the original data. Finally, the data values obtained following the translation conversion were all within the standard range interval, which determines that the $H_y$ and $H_x$ data sequences are also suitable for the GM (1,1) model's construction. In the prediction process, the K value of prediction D was 0.334, and the small-error probability $\rho$ was 0.8, which indicates that the model accuracy of prediction D is extremely high. The K value of predicting $H_y$ was 0.486, and the small-error probability $\rho$ was 0.7, which indicates that the model accuracy of predicting $H_y$ was qualified. The K value of the predicted $H_x$ was 0.603, and the small-error probability $\rho$ was 0.7, which indicates that the model accuracy of the predicted $H_x$ barely qualifies.

**Table 7.** Grey prediction results of rural e-commerce logistics and agricultural modernization in Handan.

| Year | Original D | Predicted D | Original $H_y$ | Predicted $H_y$ | Original $H_x$ | Predicted $H_x$ |
|---|---|---|---|---|---|---|
| 2010 | 0.558646 | 0.559 | 0.281873 | 0.282 | 0.345537 | 0.346 |
| 2011 | 0.614928 | 0.688 | 0.301486 | 0.351 | 0.474274 | 0.620 |
| 2012 | 0.702710 | 0.697 | 0.417866 | 0.376 | 0.583538 | 0.621 |
| 2013 | 0.738636 | 0.706 | 0.474221 | 0.402 | 0.627682 | 0.622 |
| 2014 | 0.756460 | 0.715 | 0.468280 | 0.428 | 0.699260 | 0.624 |
| 2015 | 0.794243 | 0.725 | 0.485860 | 0.455 | 0.819035 | 0.625 |
| 2016 | 0.712710 | 0.734 | 0.388293 | 0.483 | 0.664494 | 0.626 |
| 2017 | 0.716914 | 0.744 | 0.361333 | 0.510 | 0.731073 | 0.628 |
| 2018 | 0.720989 | 0.753 | 0.498969 | 0.539 | 0.541555 | 0.629 |
| 2019 | 0.767818 | 0.763 | 0.717579 | 0.568 | 0.484355 | 0.630 |
| 2020 | - | 0.773 | - | 0.597 | - | 0.632 |
| 2021 | - | 0.783 | - | 0.627 | - | 0.633 |
| 2022 | - | 0.793 | - | 0.658 | - | 0.635 |
| 2023 | - | 0.804 | - | 0.689 | - | 0.636 |
| 2024 | - | 0.814 | - | 0.721 | - | 0.637 |

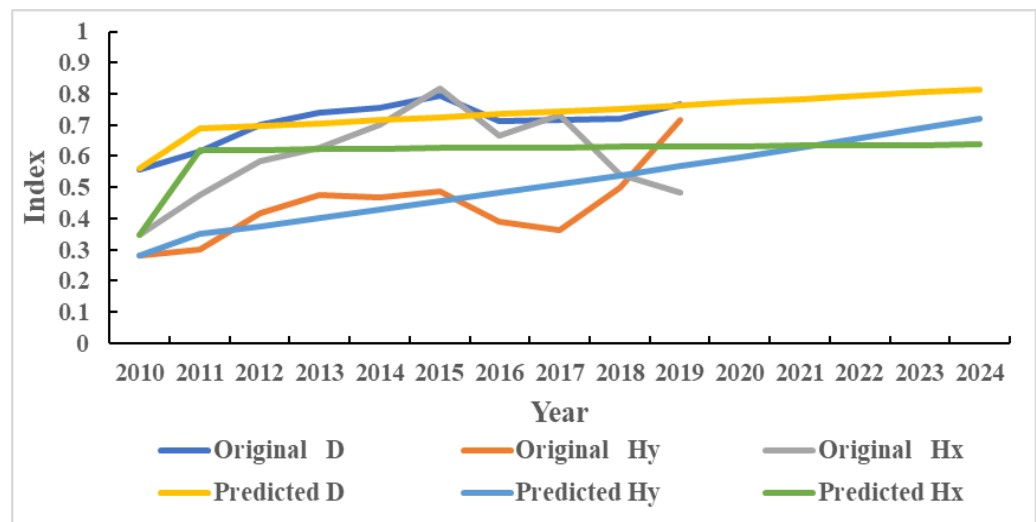

**Figure 1.** Grey prediction of rural e-commerce logistics and agricultural modernization in Handan.

From Figure 1, it can be observed that the comprehensive development level of rural e-commerce logistics and agricultural modernization in Handan City generally presents a steady upward trend from 2020 to 2024, and the gap between the two development levels will continue to narrow. It is expected to achieve a good coordination state in 2024. With the arrival of the post-pandemic era, pandemic prevention and control has become the norm. With the prosperity and development of the e-commerce industry and the government's emphasis on the rural logistics industry, rural e-commerce logistics has rapidly developed. Additionally, agricultural modernization occurred because of the vast rural area, different regional development levels, and the development is relatively slow. At the same time, under the continuous introduction of policies related to rural revitalization and e-commerce logistics in China in recent years, the coupling coordination degree of rural e-commerce logistics and agricultural modernization in Handan City will present a rapid upward trend from 2020 to 2024, which will transition from reluctant to good coordination. Therefore, if the coupling coordination degree is stable, it is necessary to consider the potential impact of the gradual development of agricultural modernization on rural e-commerce logistics, and to focus on solving the main obstacles affecting the coordinated development process, so as to improve the comprehensive development level of the two.

## 5. Discussion

Compared with previous studies, the current study has the following advantages. Firstly, a more comprehensive index system for measuring the coordinated development had been constructed from the perspectives of coordinated development, systematic conciseness, and data availability theory. The index system selected indicators from four dimensions: infrastructure, cost, efficiency, and scale, which reflected the comprehensive development level of rural e-commerce logistics. In addition, agricultural cost, efficiency, sustainable development, and social development were taken as dimensions reflecting the comprehensive development level of agricultural modernization. It provided a more comprehensive and objective measurement reference for the study of coordinated development of rural e-commerce logistics and agricultural modernization. Secondly, the model used to study the coordinated development relationship between rural e-commerce logistics and agricultural modernization, systematically analyzed the levels of comprehensive development, comprehensively evaluated the current situation of coordinated development, objectively measured the main obstacle-factors affecting coordinated development and scientifically predicted the trend of coordinated development. The quantitative evaluation of the coordinated development of rural e-commerce logistics and agricultural modernization was realized. The research results proved that the combination of entropy method, coupling coordination-obstacle degree model, and grey prediction GM (1,1) model is beneficial to scientific decision–making. Finally, in order to achieve high-quality and stable development of rural economy, on the basis of relevant analysis results, combined with the positioning of agricultural and rural development, the current study proposed some suggestions to promote the coordinated development of cross-border integration of rural e-commerce logistics and agricultural modernization.

The coordinated development of rural e-commerce logistics and agricultural modernization is of great practical significance to the high-quality development of the rural economy. According to the evaluation and trend prediction of the coordinated development of rural e-commerce logistics and agricultural modernization in Handan City from 2010 to 2019, the following suggestions were proposed.

(1) Strengthening infrastructure construction through government departments improve the rural e-commerce logistics service environment. According to the results of the obstacle degree calculation, fixed assets investment in transportation, storage and postal industries was one of the main obstacle-factors affecting rural e-commerce logistics and agricultural modernization. The government departments need to improve rural roads, the Internet, and other infrastructure constructions. First of all, older roads should be maintained, upgraded, or demolished and rebuilt; roads should be built for villages that do not have access roads; rural e-commerce logistics transportation hub sites should be systematically designed, and the "city–county–township–village" four-level logistics distribution network system should be improved to promote the development of the rural logistics industry. Secondly, the construction of the rural information network should be strengthened. Moreover, the network coverage and access rate in rural areas should be expanded, the comprehensive coverage of township broadband in Handan City should be ensured, basic coverage of a large-scale village network should be provided, the Internet should be integrated into the work and life of the villagers, and the construction process of the rural logistics system should be accelerated. In addition, the location of rural logistics information sites for professional distribution optimizes and enhances its service function, and standardizes the management of existing rural e-commerce service stations, information service stations, and logistics terminal networks, to ensure that agricultural information is consistent, accurate, and timely. Finally, governments at all levels should also introduce timely logistics equipment inspection policies, and deadlines for the recycling and centralized treatment of potentially dangerous materials or discharges that do not meet the old out-of-repair logistics vehicles. They should also be concerned with the poor health of the environment and rectify warehouse storage management, as

well as offering financial support logistics to enterprises with a large freight volume in rural areas to increase the quantity of freight-supporting equipment, because policies at the village level have high prices for logistics transportation vehicles.

(2)  Government departments should strengthen cooperation with scientific research institutions and enterprises, promoting the development of agricultural modernization through the construction of industrial and supply chains. Based on the model analyzed results, in order to achieve high-quality coordination between rural e-commerce logistics and agricultural modernization in 2024, it is necessary to focus on promoting rural industrialization development, improving rural industrial and supply chains. First of all, in the industrial chain, the related departments guide agricultural leading enterprises to drive field experts or produce high industrial production rates, establish green agricultural experimental garden for the intensive cultivation of agricultural products, and produce agricultural products through leading enterprises to integrate packaging to reduce logistics and transportation costs. We fully support the leading enterprises of rural e-commerce logistics in Handan City to use trademarks and trade names as a link; adopt franchise chain and other franchise methods; absorb rural small and micro-e-commerce logistics operators; establish modern logistics parks to alleviate the problem of the weak dispersion of rural market players; and realize the specialization and industrialization of rural e-commerce logistics. Secondly, they should strengthen R&D investment in cold-chain technology. The scientific research institutions and logistics enterprises in Handan City should strengthen their cooperation, expand the research team of cold-chain logistics technology, and jointly develop and produce core technology and equipment involved in of cold-chain logistics transportation. Local government finance helps rural e-commerce logistics enterprises to install advanced and applicable cold-chain logistics transportation and storage equipment. E-commerce logistics enterprises design the overall planning methods, scientifically allocate cold-chain transportation vehicles and storage warehouses, and improve freight efficiency. Finally, under the goal of "double carbon", the Handan government should vigorously promote green food and organic agricultural products, promote the standardization of agricultural production, increase the added value of agricultural products, establish special high-quality agricultural areas in central and southern Hebei, form a green agricultural supply chain, optimize logistics and transportation costs, and promote agricultural modernization.

(3)  The science and technology department should increase its research efforts and strengthen cooperation with the agriculture and transport departments, leading the coordinated development of rural e-commerce logistics and agricultural modernization with big data technology. According to the relevant analyzed results, rural freight volume, total rural foreign-trade export volume, and rural per capita net income were the main obstacle-factors affecting rural e-commerce logistics and agricultural modernization from 2010 to 2019. The logistics and transportation process needs to maintain information management through big data technology. In addition, the corresponding increase in logistics jobs is needed to promote the development of logistics and agriculture while improving farmers' income. First of all, the transportation department combines urban and rural public transportation systems with the logistics transportation system, and uses advanced logistics technology to improve the rural e-commerce transaction and freight volumes, so as to realize the interconnection and sharing of logistics and passenger flow. Secondly, the science and technology department promotes and popularizes RFID and other network-sensing technologies in rural areas. Through the Beidou navigation and intelligent logistics system, it comprehensively monitors, continuously tracks, and feeds back the problems in all aspects of e-commerce logistics in a timely manner. With the help of the Internet of Things, artificial intelligence, and "cloud computing", it promotes the automation and intelligent operation of logistics equipment and strengthens the connection between the main bodies in the supply chain. Thirdly, the local government establishes and

improves the public information-sharing platform at county and township levels, builds a comprehensive resource-exchange information-sharing platform integrating agriculture, agricultural trade, logistics, and e-commerce management, and the "media convergence" agricultural information service platform. With the help of big data technology, the process of agricultural product logistics and transportation is transparent and traceable. Finally, in remote rural areas or areas with traffic congestion, Handan City relies on the basic realization of rural Internet full-coverage advantages, expanding to create a "crowdsourcing logistics" model to increase employment rates, to attract rural residents to participate in express delivery businesses, ensure that the villagers express deliveries do not cease, and create a variety of resource interconnections, reduce the cost of e-commerce logistics, logistics, and freight speed to solve the rural logistics "last mile" problem, in order for Handan City to achieve rural e-commerce logistics and agricultural modernization coordinated development.

## 6. Conclusions

With industrial integration becoming a global industrial phenomenon and the arrival of the post-pandemic era, the coordinated development of rural e-commerce logistics and agricultural modernization has become an important method to achieve healthy regional economic development. As a basic and emerging industry in the development of the national economy, the agriculture and e-commerce logistics industry achieves cross-border integration, providing a successful example for the integration of three industries in China. The cross-integration and coordinated development of rural e-commerce logistics and agricultural modernization is conducive to promoting the prosperity of the e-commerce industry. It is also conducive to the precise docking of the supply and demand of agricultural products, the close integration of production and marketing, the promotion of rural informatization, the acceleration of urban–rural integration, and the sustainable development of the regional economy. The innovations are as followings.

(1)  The current study constructed the measurement index of the coordinated development of rural e-commerce logistics and agricultural modernization. The index system is constructed from the perspectives of coordinated development, system simplicity, and data availability, which provides a more comprehensive and objective measurement reference for the coordinated development of rural e-commerce logistics and agricultural modernization.

(2)  The current study combined with the entropy method, the coordinated development of rural e-commerce logistics and agricultural modernization was quantitatively evaluated and trend predicted by coupling coordination-obstacle degree model, and grey prediction GM (1,1) model. The research process was more objective and systematic, and the evaluation results were supported by data. This method is helpful to analyze the coordinated development status of rural e-commerce logistics and agricultural modernization, find out the obstacle factors, and provide data support for promoting the coordinated development of the two.

(3)  The case results show that the method is feasible. Through the case study, the coordinated development relationship between the system can be objectively and scientifically measured and evaluated, and targeted development suggestions can be put forward, which will play a positive role in promoting the coordinated development of regional rural e-commerce logistics and agricultural modernization, and boost the stable development of the regional economy.

Beginning with the concept of coordinated development, this paper studies the coordinated development of rural e-commerce logistics and agricultural modernization in Handan. This paper also analyzed the coupling coordination degree of the two principles from 2010 to 2019, and the obstacle-factors affecting the coordinated development, and predicted the comprehensive development index and coupling coordination trend from 2020 to 2024. The conclusions for the coordinated development of rural e-commerce logistics and agricultural modernization were as follows:

(1) The measurement index system of rural e-commerce logistics and agricultural modernization was constructed. In the construction of the measurement index system, based on the concept of coordinated development, considering the development of information technology and agricultural industrialization, this paper focuses on the selection of indicators such as rural mobile-phone users and the agricultural industrialization management rate, and constructed the measurement index system for the coordinated development of rural e-commerce logistics and agricultural modernization from the system and index levels.

(2) The coupling coordination-obstacle degree and grey prediction GM (1,1) models were established. The current study comprehensively used the coupling coordination and obstacle theory to establish a coupling coordination-obstacle degree model. The calculation results show that the total amount of rural postal and telecommunication services and the per capita net income of farmers are the main obstacles to the coordinated development of rural e-commerce logistics and agricultural modernization. The grey prediction GM (1,1) model was introduced in the prediction of the coordinated development of regional rural e-commerce logistics and agricultural modernization, and the trend of the coordinated development of rural e-commerce logistics and agricultural modernization was analyzed. It was expected that the two will achieve a good coordination state in 2024. The model was feasible.

(3) We proposed the coordinated development of rural e-commerce logistics and agricultural modernization. Based on the concept of coordinated development, from the perspective of relevant government departments, we proposed the strengthening of the regional rural infrastructure to provide hardware support for the development of rural e-commerce logistics. We used the construction of industrial and supply chains to promote the development of agricultural modernization, and used big data technology to lead the coordinated development of rural e-commerce logistics and agricultural modernization.

However, regional rural e-commerce logistics and agricultural modernization is a complex system of mutual influence and dynamic development. Under the background of deepening industrial integration, the synergy between rural e-commerce logistics and agricultural modernization is becoming increasingly close. There are still several shortcomings evident in the current study. For example, the index system of evaluation needs to be dynamically optimized according to social development. Furthermore, the research can explore the path of the coordinated development of rural e-commerce logistics and agricultural modernization under artificial intelligence technology. In addition, more intelligent analysis methods can be proposed to promote the high-quality development of regional rural e-commerce logistics and agricultural modernization.

**Author Contributions:** Conceptualization, Z.L. and C.G.; data curation, S.J.; formal analysis, Y.N. and Z.W.; investigation, S.J. and Y.N.; methodology, S.J. and Z.L.; supervision, C.G.; validation, Z.L.; writing—original draft, Z.L.; writing—review and editing, C.G., Z.W. and S.J. All authors have read and agreed to the published version of the manuscript.

**Funding:** This research was funded by the Social Science Fund of Hebei Province of China (HB18GL023), the Major Program of Humanities and Social Sciences Research of Education Department of Hebei Province of China (ZD201815), and the S&T Program of Hebei Province of China (21557682D).

**Institutional Review Board Statement:** Not applicable.

**Informed Consent Statement:** Not applicable.

**Data Availability Statement:** Data are available upon request from the corresponding author.

**Conflicts of Interest:** The authors declare that they have no conflict of interest.

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
