# Peer review of "A Measurement Model and Empirical Analysis of the Coordinated Development of Rural E-Commerce Logistics and Agricultural Modernization"

_sustainability, doi:10.3390/su142113758_

Round 1

Reviewer 1 Report

Please see attached report. 

Reviewer 2 Report

A Measurement Model and Empirical Analysis of Coordinated Development of Rural E-commerce Logistics and Agricultural Modernization. This research applied  Grey prediction GM (1,1) model to forecasting and found that the coordinated development of rural e-commerce logistics and agricultural modernization will achieve good coordination in 2024. The research build strategy obstacle factors analysis, and trend prediction of coordinated development of regional rural e-commerce logistics and agricultural modernization in Handan City, Hebei Province China. After reading this paper, I have some suggestions for the author as follows:

1. Still have some errors typing Ex: Lines 55~57; 62~63, 66, 96~97. 379.

2. The literature review of the research topic is too weak, mainly listing issues without sound analysis and evaluation.

3. Grammar and sentence structure.

4. The reference format needs a double check, it is still a mistake.

5. What is the difference between e-commerce logistics and Agriculture Logistics?

6. What does it mean "In order to avoid the data of a standardized index is 0 and affect the subsequent calculation" the author mention?

7. Why the author decided to do steps (1) and (2) before doing forecasting? Please explain clearly how to calculate it. Which research did the author citation it?

8. Why authors integrated entropy method, system comprehensive development index, system coupling coordination degree, obstacle degree, and Grey (1,1). Please explain it. Cause this step is an implication of this research. What is the advantage if the author applied it? Normally in the previous research, they just apply Grey (1,1) to forecasting. 

Round 2

Reviewer 2 Report

ð 1. The reference format needs a double check, it is still have some mistake.

ð 2. There is no clear explanation give about the data used in the model - include a data section with clear reference made to the source of the data and the period covered. Please explain each step of the Formulas (1) - (9) in this research "4.2. The calculation of coupling coordination-obstacle degree" and how to calculate each step, Cause this is the implication in your research.

ð I hope the authors can improve this research

Round 3

Reviewer 2 Report

Dear Author,

Good luck to your team. Thanks for your contribution to this research.